# A Neuroevolutionary Model to Estimate the Tensile Strength of Manufactured Parts Made by 3D Printing

**Matheus Alencar da Silva** [1]**, Bonfim Amaro Junior** [2]**, Ramon Rudá Brito Medeiros** [1,3] **and Plácido Rogério Pinheiro** [4,*]

[1] Undergraduate Mechanical Engineering Program, Federal University of Ceará, Russas 62900-000, Brazil; matheusalencar6942@gmail.com (M.A.d.S.); ramon.ruda@ufc.br (R.R.B.M.)

[2] Núcleo de Estudo em Machine Learning e Otimização (NEMO), Federal University of Ceará, Russas 62900-000, Brazil; bonfimamaro@ufc.br

[3] Graduate Program in Mechanical Engineering, Federal University of Paraíba (UFPB), João Pessoa 58051-900, Brazil

[4] Graduate Program in Applied Informatics, University of Fortaleza (UNIFOR), Fortaleza 60811-905, Brazil

[*] Correspondence: placido@unifor.br

**Abstract:** Three-dimensional printing has advantages, such as an excellent flexibility in producing parts from the digital model, enabling the fabrication of different geometries that are both simple or complex, using low-cost materials and generating little residue. Many technologies have gained space, highlighting the artificial intelligence (AI), which has several applications in different areas of knowledge and can be defined as any technology that allows a system to demonstrate human intelligence. In this context, machine learning uses artificial intelligence to develop computational techniques, aiming to build knowledge automatically. This system is responsible for making decisions based on experiences accumulated through successful solutions. Thus, this work aims to develop a neuroevolutionary model using artificial intelligence techniques, specifically neural networks and genetic algorithms, to predict the tensile strength in materials manufactured by fused filament fabrication (FFF)-type 3D printing. We consider the collection and construction of a database on three-dimensional instances to reach our objective. To train our model, we adopted some parameters. The model algorithm was developed in the *Python* programming language. After analyzing the data and graphics generated by the execution of the tests, we present that the model outperformed, with a determination coefficient superior to 90%, resulting in a high rate of assertiveness.

**Keywords:** fused deposition modeling; 3D printing; artificial intelligence

## 1. Introduction

Humanity has undergone several technological advances, and is currently experiencing the fourth industrial revolution, with effects in several areas. New technologies are constantly being developed, always requiring new knowledge. One of them is 3D printing, which still undergoes constant evolution [1]. Three-dimensional printing is a manufacturing area that builds parts, starting from a virtual model, by the automatic layer-by-layer deposition method [2].

In the late 1980s, 3D printing had a high cost and detailed coverage. In the 2000s, mainly due to the fall in patents, this process became less expensive, allowing for applications in other areas, such as education and medicine. In addition to having become a technology with several applications, 3D printing has gained strength by using low-cost products and generating little waste [1].

There are a few 3D printing methods, the most common being fused filament fabrication (FFF). This method consists of the hot deposition of a filament through the extrusion of a specific material [3].

Despite being a widely used process, 3D printing is a very elaborate technology. This high complexity is due to the many parameters influencing the process and the final effect. Geometry, speed, temperature, raw material and other factors influence the final properties of the product, such as mechanical strength and dimensional accuracy. Several works have already been developed to define how each 3D printing parameter influences the outcome. However, even with this large amount of studies, there are some difficulties in developing mathematical models that describe the final properties of the products [2].

Some works applied a computational strategy known as an artificial neural network (ANN) to control and optimize the parameters of the 3D printing process in the best possible way. Neural networks can present a supervised, unsupervised and reinforcement learning application methodology, being part of an even broader branch that is known as artificial intelligence (AI) [4].

The ANN represents an information processing system that simulates the functions of the human brain computationally. Neural networks, as well as other supervised learning methods, start from a data set, with input values associated with their respective output, so, after executing an algorithm, known as a learning algorithm, the network will have the ability to classify or predict the outcome of cases where the inputs do not have a known output [5]. The aforementioned concept can be beneficial for several applications, but these types of methods have several parameters that can be considered. Several tests can be carried out to define the best configuration for the ANN; however, performing this task can take a long time. Several methods can be used to mitigate the situation above. This paper applies a genetic algorithm (GA), commonly used to tackle optimization problems and grounded on Darwin's theory of evolution, addressing steps such as natural selection, reproduction and mutation [6]. Combining ANN with the GA is called the neuroevolutionary strategy [7].

In this context, our paper aims to develop a model that combines artificial neural networks and genetic algorithms in a neuroevolutionary strategy, capable of predicting the rupture stress of materials manufactured by the 3D printing method of the FFF type. The main idea is to realize the training procedure of an ANN using the genetic algorithm strategy and to validate the weights representation throughout as a feasible solution. We highlight that models applying neural networks are commonly different in the configurations and representation of weights, activation functions and learning methods. Therefore, in this paper, we focus on presenting the choice composition of all of the above characteristics to obtain our proposal in the 3D printing process.

The remaining part of this paper is organized as follows. Section 2 reviews the literature and theoretical fundamentals considering 3D printing (Section 2.1), ANN and essential aspects about training and test sets applied in models that use ANNs in their conception (Section 2.2), GA (Section 2.3). Next, Section 3 details the implementation of a neuroevolutionary model to estimate the tensile strength of manufactured parts made by 3D printing. Section 4 presents the used datasets and conferred results for this research. Section 5 concludes by summarizing the results and limitations of the presented approach and comments on future research directions.

## 2. Fundamental Concepts

This section presents the fundamental concepts of our neuroevolutionary model definition. The main objective is to estimate the tensile strength of manufactured parts made by 3D printing. Then, we introduce this technology in Section 2.1. In addition, aspects related to the ANNs and notions about models data analysis (Section 2.2) and GA (Section 2.3) are presented.

### 2.1. Three-Dimensional Printing

The world is constantly witnessing several technological advances. These advances always require acquiring new knowledge so that they are better understood. A technology that has evolved significantly in recent decades is 3D printing [1].

One of the significant advantages of this technology is the flexibility in producing 3D-modeled parts by computer-aided design (CAD), which, together with computer-aided manufacturing (CAM), allows for the fabrication of different geometries, complex or straightforward [8]. Additive manufacturing technology can be divided into seven families: vat photopolymerization; powder bed fusion (PBF); binder jetting (BJ); material jetting (MJ); sheet lamination (SL); material extrusion (ME); directed energy deposition (DED) [9]. In the material extrusion (ME) family, the FFF process is the most common and easily used nowadays. They are the printers that generally anyone can have at home due to their low cost and operational simplicity [10].

In the FFF process, the raw material, in the form of a filament, is hot extruded through a nozzle on a heated table following the coordinates defined by the digital file, where the material is deposited layer by layer until the piece is obtained [2]; see Figure 1. The most common materials in this type of process are plastic filaments, such as PLA, which is a rigid material, which allows fir greater detail in the parts produced with it, and acrylonitrile butadiene styrene (ABS), which is a thermoplastic with great flexibility and has more excellent resistance to impacts. Other materials also used in this process are polyester, polypropylene (PP), polycarbonate (PC), polyamide (PA), elastomers and waxes [3,11].

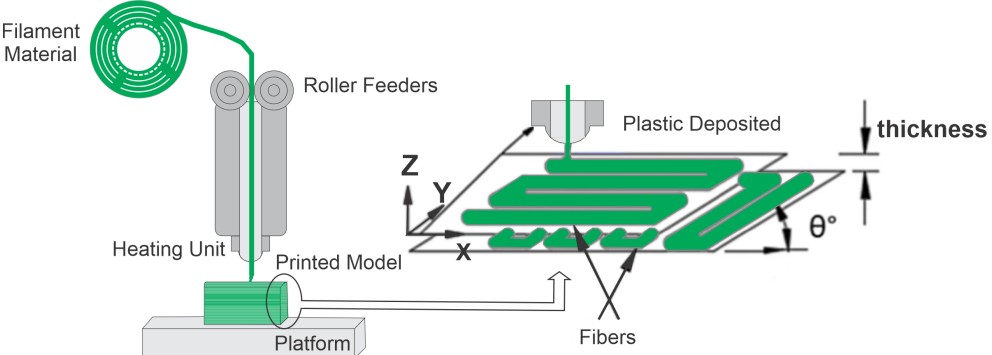

**Figure 1.** Three-dimensional printing process.

In the 3D printing process, several parameters influence the manufactured product [12]. Therefore, to have greater control over the final part, each of these factors must be carefully analyzed. Some of these parameters are [2,13]:

- Construction orientation: orientation with which the part is constructed relative to the base, along the X, Y and Z axes.
- Layer thickness: height of the layer deposited by the extruder nozzle. Varies with nozzle diameter and material.
- Fill density: space between adjacent filaments in the fill region of the part.
- Fill angle: angle between the filler filament deposition and the X axis.
- Filler filament width: width of the filament used for filling. It depends on the diameter of the extruder nozzle.
- Number of contours: number of perimeters built along the part, internal and external.
- Contour filament width: width of the filament used in the contour of the part.
- Space between contour filaments: space between each of the filaments used in the contour.
- Space between contour and fill: space between the contour and the effective fill of the part.
- Extrusion speed: speed at which extruded filaments are deposited.
- Extrusion temperature: temperature at which the filament is deposited.
- Platform temperature or bed temperature: temperature of the surface of the table/bed where the printing is carried out.
- Environmental conditions: temperature and humidity of the printing environment.

Due to this large number of factors that can influence the manufacture of parts produced by 3D printing, physical and mathematical formulations become challenging to develop [14].

*2.2. Artificial Neural Networks*

Artificial neural networks are a computational representation of an information processing system with characteristics similar to biological neural networks, inspired by the functioning of the human brain [15].

The artificial neuron is the basic processing unit of an ANN, its first model being proposed by McCulloch and Pitts in 1943 . In the model in question, the dendrites are represented by $n$ inputs $x_1$, $x_2$,...$x_n$. The axon is characterized by the $y$ output. The synapses (connections between neurons) are responsible for defining the intensity of each input signal, being represented by the synaptic weights $w_1$, $w_2$,...$w_n$, each one associated with its respective input. The nucleus and the cell body are responsible for handling the inputs and calculating the weighted sum $u$ of the inputs, in addition to verifying if this sum exceeds the threshold $\theta$; if it exceeds, the neuron fires a $y$ signal. Otherwise, no signal is fired. Some methods based on the McCulloch and Pitts model allow for different output signals. For this result, different activation functions were defined, such as linear, sigmoid, hyperbolic tangent, inverse tangent and ReLU, among others [16].

There are some types of networks, the most common being perceptron, Kohonen, Hopfield and ART [17]. In this paper, we applied the perceptron type ANN. The basic unit of this type of network is the simple perceptron, which works by receiving a set of input data and a bias, weighted by their respective synaptic weights. Then, the sum of these data is calculated, and then the activation function is triggered; the result sorts the input set between two different groups.

However, the simple perceptron has some limitations. This type of network has problems classifying sets that are not linearly separable, or even when this separation is not well defined. For this type of situation, it is more appropriate to use the multilayer perceptron (MLP) network.

MLP consists of an input layer, an output layer and one or more hidden layers between these two layers. Layers are intended to increase the network's ability to model complex functions. Each layer in a network contains a sufficient number of neurons depending on the application. The input layer is passive and works by just receiving the data. The hidden and output layers actively process the data, the output layer being responsible for producing the results of the neural network [18]. The MLP model has supervised learning by error correction, has more than one layer and is acyclic. The output of a neuron cannot serve as an input for any last neuron that is connected. Therefore, all neurons process each input. A propagation rule is given by the inner product of the inputs weighted by the weights with the addition of the bias term, and the output of the previous layer is the input of the current layer. It is important to note that, in this type of network, there may be more than one hidden layer, in addition to different numbers of neurons in each layer [5].

Only a training set and arbitrary synaptic weights are not enough for a neural network to classify or predict values closest to an actual situation. For this, it is necessary to carry out training. The weights are adjusted, better describing the condition addressed. The training of an MLP network is usually divided into a few steps, called the feedforward step (forward phase) and the backpropagation with the adjustment of the weights (backward phase) [19].

The backpropagation algorithm is one of the most used tools for ANN training. However, in some practical applications, it may be too slow.

Suppose, for example, that there are $t$ training samples, $f$ features and $h$ hidden layers, each containing $n$ neurons and $o$ output neurons. The time complexity of backpropagation is $O(t * f * n^h * o * i)$, where $i$ is the number of iterations. Since backpropagation has a high time complexity, it is advisable to start with a smaller number of hidden neurons and a few hidden layers for training. Note that *Big-O* quota is a mathematical notation that describes

the limiting behavior of a function when the argument tends towards a particular value or infinity.

Even with a finalized optimization, ANNs can run into other problems, such as underfitting and overfitting. These two errors are more commonly seen in the construction of neural networks. They are the trend error and the variance error. The bias error arises because the network tries to describe a generalized behavior for its data, which does not suit the noise sufficiently and dictates a simplified trend.

The network is tied to noise in the variance error and generates an excessively complex model. Combining a high trend error with a low variance error generates underfitting, where the developed model is straightforward, not fitting the points and, consequently, not correctly describing the natural phenomenon. When the opposite happens—that is, the model has a low tendency and high variance—overfitting is generated, where the network adapts excessively to the training data and loses the ability to generalize to points outside of the training set [17].

The neural network can be evaluated by its ability to fit the training data and predict data outside this set. The available data are generally divided into training, validation and testing to improve the model results. During the training stage, the outcomes calculated by the network are compared with the target (supervised learning); then, the weights are adjusted to approximate them. Then, in the validation step, the model undergoes a fine adjustment in the parameters, avoiding rigidity to the training data, thus reducing the chances of overfitting. Finally, the prediction is performed with the data that were separated for the test step, and then the result is compared with the absolute values [20].

Underfitting represents when the model performs poorly on training and test data. In contrast, overfitting indicates that the model acted well on the training data but then struggled on the unrecognized inputs; this should not be the case, as both data groups came from the identical distribution. One of the ways to evaluate the model's performance is the study of the learning curves. An analysis of the model with the training and test data can diagnose the possibilities of overfitting and underfitting [21].

Many techniques have been explored to accelerate its performance, considering that it may fall into local minima. One of the potential treatments to escape from local minima is by operating a minimum learning rate, which slows down the learning process. In [22], the authors present a new strategy based on the use of the bi-hyperbolic function, which offers greater flexibility and a faster evaluation time. On the other hand, we applied the genetic algorithm to the training of our neural networks.

### 2.3. Genetic Algorithm

The genetic algorithm (GA) is widely used in optimization, which simulates Darwin's theory of evolution. GA differs from other methods in three main points:

- This method works from a population of solutions to the problem.
- This method does not depend on differential equations.
- This method uses probabilistic and non-deterministic rules.

The GA starts from a set of possible solutions to the problem addressed. This set is known as the population. Each individual in this population is characterized by the chromosome, the set of values that solve the problem. Each of these values is known as a gene. Each gene can be encoded in different ways, such as in binary, integer, double precision or other ways [23,24]. Figure 2 presents the GA process flowchart.

The zero-step process is the generation of a population with an amount $N$ of individuals, where each one of them has $i$ genes. The value of each gene must be generated randomly, and its encoding depends on the problem at hand [24].

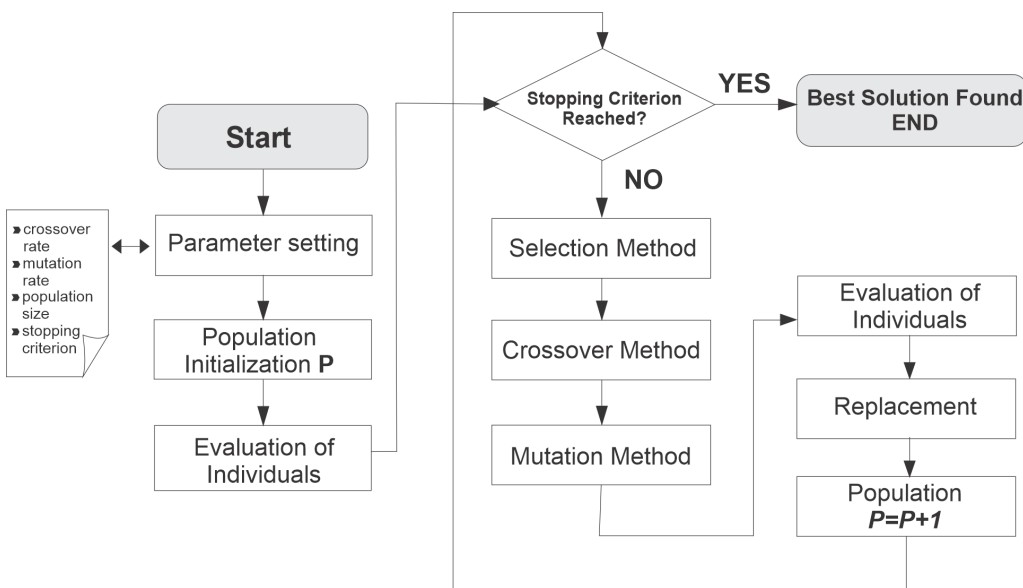

**Figure 2.** The standard workflow of Genetic Algorithm metaheuristic.

After the initial population is generated, the evolutionary process begins. The first step is the aptitude assessment or calculation. At this point, the adaptation degree of each individual is calculated. As each solution set improves, it tends to the desired response, thus offering the GA an aptitude measure of each individual in the population. Together with the chromosome coding, this step is the most dependent on the problem addressed, varying for each case. The choice of the fitness function is critical to the success of the algorithm [6].

Then comes the crossing or reproduction step (crossover). This process creates more fit populations, with better solutions, from the individuals selected in the previous step. In this method, the individual's temporary sets are separated into pairs, and if a specific number (crossover rate), randomly generated between 0 and 1 for each pair, is greater than the probability of mating, then the pair, known as parents, goes through the process of reproduction, giving rise to two new individuals, the children, who will make up the new population [24].

Behind the crossover step, the mutation process takes place. The mutation operator is necessary for introducing and maintaining the population's genetic diversity, arbitrarily altering one or more components of a chosen structure, thus providing means for introducing new elements into the population. In this way, mutation ensures that the probability of reaching any point in the search space will never be zero, in addition to circumventing the problem of local minimums [6].

In most GA applications, all individuals in a population are replaced by new ones, but some of the previous population is propagated to the next generation. That is, not the entire set is renewed. There are also cases where the population size varies according to the generation. At the end of each generation, a population is generated with individuals that are, for the most part, more fit than the individuals of the previous generation [25].

There are some stopping criteria for the genetic algorithm. The main ones are the number of generations or the degree of convergence of the current population. After each generation, the population passes an evaluation, and if the pre-established criterion is fulfilled, the algorithm stops. At the end of the algorithm's execution, the population will contain the individuals that best fit as a solution to the problem studied, thus optimizing the practical case, be it maximization or minimization.

## 3. The Neuroevolutionary Model

The model applied in this paper followed sequential steps. Initially, data related to 3D printing were collected, and both parameters of the printing process and properties of

products manufactured by the studied process were considered. It is crucial to analyze the data to establish a consistent basis. Then, a genetic algorithm was elaborated on, defining a configuration to optimize the parameters of the ANN, with the ANN being the target of the next step. The neural network was designed to adapt to the data collected in the first step. The next step consisted of several executions of the combined algorithms to generate enough data to follow the workflow of the process adopted.

The first step consisted of collecting experimental data from works that measured the mechanical properties of 3D-printed products made from polylactic acid (PLA). The extracted data were printing speed, extrusion temperature, fill density, extruded filament thickness, extrusion orientation and tensile strength. Data from seven works were selected, totalling 149 input–output sets, which were named as an instance; these works are listed in Table 1. The complete database used is shown in Table A1. Table 2 presents a range of values for the properties used, showing a good range for each parameter.

**Table 1.** Works used.

| Work | Author(s) |
|---|---|
| Effect of print speed and extrusion temperature on properties of 3D printed PLA using fused deposition modeling process | [26] |
| Effect of process parameters on mechanical properties of 3D printed PLA lattice structures | [27] |
| Effects of fused deposition modeling process parameters on tensile, dynamic mechanical properties of 3D printed polylactic acid materials | [28] |
| Mechanical Properties on ABS/PLA Materials for Geospatial Imaging Printed Product using 3D Printer Technology | [29] |
| Tensile failure strength and separation angle of FDM 3D printing PLA material: Experimental and theoretical analyses | [30] |
| Estudo da Influência de Parâmetros de Impressão 3D nas Propriedades Mecânicas do PLA | [31] |
| Influência dos parâmetros de impressão 3D na resistência à tração de corpos de prova impressos em PLA utilizando modelagem por fusão e deposição | [32] |

To prepare the genetic algorithm, he computational structure of the individual was defined. A configuration with 118 genes was chosen, which is necessary to represent all of the parameters adopted in the neural network; see Figure 3. The chosen representation was binary, as it adequately met the need for this proposal. Of the individual's 118 positions, the first 25 represent the value of the training rate ($\gamma$). The genes at positions 26 to 50, 51 to 75 and 76 to 100 are the values of the hyperparameters ($\beta_1$ and $\beta_2$) and numerical stability ($\epsilon$), respectively. The genes at positions 101 to 106, 107 to 112, and 113 to 118 represent the values of neurons in each of the three hidden layers. A representation of the individual used in the genetic algorithm is shown in Figure 3. To evaluate the training quality of each one of the networks, the determination coefficient $R^2$ was chosen. This metric is operated to examine how a difference in a second variable can describe disparities in one variable. The benefit of $R^2$ is its power to find the possibility of future events falling within the predicted outcomes [33].

**Table 2.** Range of values of the properties used.

| Property | Minimum | Maximum |
|---|---|---|
| Print speed (mm/s) | 30 | 80 |
| Extrusion temperature (°C) | 185 | 240 |
| Fill density (%) | 10 | 100 |
| Thickness (mm) | 0.05 | 0.3 |
| Orientation (°) | 0 | 90 |
| Tensile strength (MPA) | 17.67 | 59.84 |

We adopted uniform random initialization to generate the initial population and considered 100, 500 and 1000 as possible values in our execution tests to verify how the population size influences the model quality and complexity.

For the selection method, the tournament with a number of individuals equal to three was defined [34]. In the crossing step, the multipoint method with two cut-off points was selected. In the mutation step, the flip method was chosen, with crossover and mutation probability values of 90% and 5%, respectively. The maximum number of generations adopted was 1000, but the algorithm must stop its execution whenever the best individual of each generation remains the same for ten consecutive generations. A time limit of three hours was established for each run.

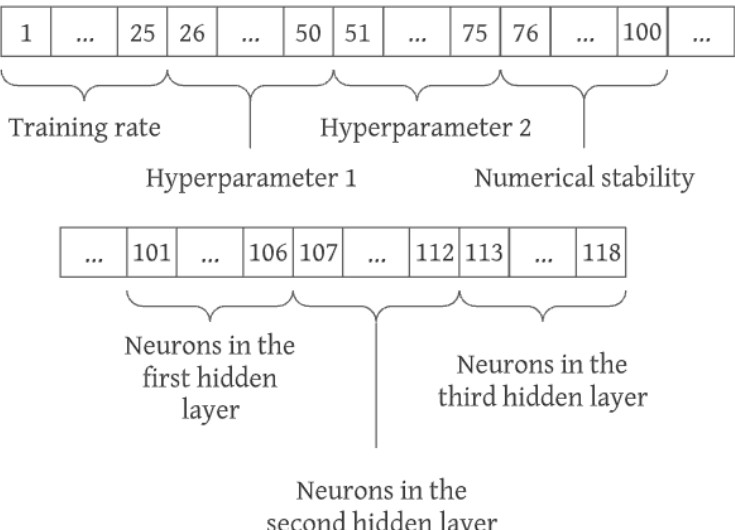

**Figure 3.** Illustration of the individual representation used in genetic algorithm proposed.

The type of neural network adopted was the multilayer perceptron with the application of the Adam method. The maximum number of 100 iterations in the network proved to be satisfactory. The deactivation function chosen was ReLU. For the number of hidden layers, the amount of three proved to be interesting, as it converged to good results in reasonable time intervals. To define a data distribution between the training, validation and test groups, the algorithm was executed several times with different values [21]. Table 3 shows the adopted distributions. Note that we divided the sets into three. The objective is to mitigate the possible effects of underfitting or overfitting and obtain good results even when applied to new untested instances.

**Table 3.** Data distribution between training, validation and testing.

| Distribution | Training | Validation | Testing |
|:---:|:---:|:---:|:---:|
| 1 | 60% | 10% | 30% |
| 2 | 70% | 10% | 20% |
| 3 | 80% | 10% | 10% |

## 4. Results and Discussion

The algorithms were developed in the Python programming language using the following libraries: Pandas for reading data from comma separated values (CSV) files, which are file types used to organize data, Numpy for manipulating the data, Matplotlib for generating graphs, Scikit-learn for implementing the ANN and Pygad for implementing the genetic algorithm.

At each run, variations were performed in the population size of the genetic algorithm and the sizes of the training, validation and testing sets of the neural network. In addition, each of the variations were performed ten times to acquire more data for further analysis. The variations adopted can be seen in Table 4.

**Table 4.** Variations in the developed algorithms.

| Variation | Training | Validation | Testing | Population |
|:---:|:---:|:---:|:---:|:---:|
| V1.1 | 60% | 10% | 30% | 100 |
| V1.2 | 60% | 10% | 30% | 500 |
| V1.3 | 60% | 10% | 30% | 1000 |
| V2.1 | 70% | 10% | 20% | 100 |
| V2.2 | 70% | 10% | 20% | 500 |
| V2.3 | 70% | 10% | 20% | 10,000 |
| V3.1 | 80% | 10% | 10% | 100 |
| V3.2 | 80% | 10% | 10% | 500 |
| V3.3 | 80% | 10% | 10% | 1000 |

The different configurations generated by the algorithm's execution underwent a detailed analysis. Observing the $R^2$ coefficients and the learning curves, the configuration that best applies to the situation studied was then defined. The coefficient of determination, also called $R^2$, is a measure of fit of a generalized linear statistical model to the observed values of a random variable. $R^2$ varies between 0 and 1, and is sometimes expressed in percentage terms. This case expresses the amount of data variance that the linear model explains. Thus, the higher the $R^2$, the more explanatory the linear model is, and the better it fits the sample. This decision is made grounded on the best coefficients, not overfitting or underfitting. The algorithm itself calculated the $R^2$ values. Table 5 shows the determination coefficient value for each run performed, along with the mean, standard deviation and average time of the computing performance. Note that v3.2 presented the best average among all during the execution of the selected variations. These data are graphically represented in Figure 4, considering the x-axis to analyze the variations in the dataset division parameters to generate the model, and the y-axis to expose the average (points) and the deviations (vertical lines in the same point).

**Table 5.** Determination coefficient $R^2$.

| Execution | V1.1 | V1.2 | V1.3 | V2.1 | V2.2 | V2.3 | V3.1 | V3.2 | V3.3 |
|---|---|---|---|---|---|---|---|---|---|
| 1 | 0.8427 | 0.8644 | 0.8981 | 0.9392 | 0.9439 | 0.9572 | 0.9435 | 0.9799 | 0.9705 |
| 2 | 0.8593 | 0.9180 | 0.9000 | 0.9461 | 0.9542 | 0.9522 | 0.9693 | 0.9508 | 0.9815 |
| 3 | 0.8894 | 0.8801 | 0.9080 | 0.9442 | 0.9544 | 0.9469 | 0.9571 | 0.9833 | 0.9831 |
| 4 | 0.8935 | 0.8994 | 0.9132 | 0.9490 | 0.9521 | 0.9551 | 0.9724 | 0.9840 | 0.9860 |
| 5 | 0.8444 | 0.9057 | 0.8736 | 0.9362 | 0.9558 | 0.9526 | 0.9671 | 0.9836 | 0.9612 |
| 6 | 0.8508 | 0.9111 | 0.9164 | 0.9389 | 0.9531 | 0.9560 | 0.9444 | 0.9791 | 0.9767 |
| 7 | 0.8343 | 0.9073 | 0.9241 | 0.9412 | 0.9585 | 0.9600 | 0.9520 | 0.9803 | 0.9508 |
| 8 | 0.8364 | 0.8983 | 0.9142 | 0.9496 | 0.9600 | 0.9559 | 0.9762 | 0.9682 | 0.9776 |
| 9 | 0.8933 | 0.8873 | 0.8945 | 0.9523 | 0.9433 | 0.9573 | 0.9458 | 0.9785 | 0.9713 |
| 10 | 0.8955 | 0.9018 | 0.8928 | 0.9333 | 0.9531 | 0.9463 | 0.9480 | 0.9821 | 0.9864 |
| Mean | 0.8551 | 0.9006 | 0.9034 | 0.9427 | 0.9537 | 0.9555 | 0.9545 | 0.9801 | 0.9766 |
| Deviation | 0.0259 | 0.0159 | 0.0147 | 0.0063 | 0.0055 | 0.0045 | 0.0126 | 0.0103 | 0.0114 |
| Time(s) | 1357 | 5322 | 12,050 | 841 | 5097 | 12,135 | 726 | 5811 | 9223 |

Even with the calculated $R^2$ coefficient, it is still necessary to verify the occurrence of overfitting and underfitting. The graphs in Figures A1–A9 represent the loss curves for variations v1.1, v1.2, v1.3, v2.1 , v2.2, v2.3 , v.3.1 , v3.2 and v3.3, respectively.

**Table 6.** Values of selected parameters.

| Variable | Value |
|---|---|
| Training rate ($\gamma$) | 0.0039 |
| Hyperparameter 1 ($\beta_1$) | 0.7892 |
| Hyperparameter 2 ($\beta_2$) | 0.5625 |
| Numerical stability ($\epsilon$) | $2 \times 10^{-6}$ |
| Hidden neurons in layer 1 | 35 |
| Hidden neurons in layer 2 | 63 |
| Hidden neurons in layer 3 | 56 |

The graphs were used to check for anomalies such as overfitting or underfitting. As can be seen in Figures A4 and A6, the curves for variations 2.1 and 2.3 showed many oscillations, which may indicate the presence of anomalies. The rest of the graphs, Figures A1–A3, A5 and A7–A9, showed curves with a satisfactory behavior. Based on these graphs, on the assertiveness of each model and on the time spent for optimization, the configuration adopted is presented by variation 2.3, and the parameters are presented in Table 6.

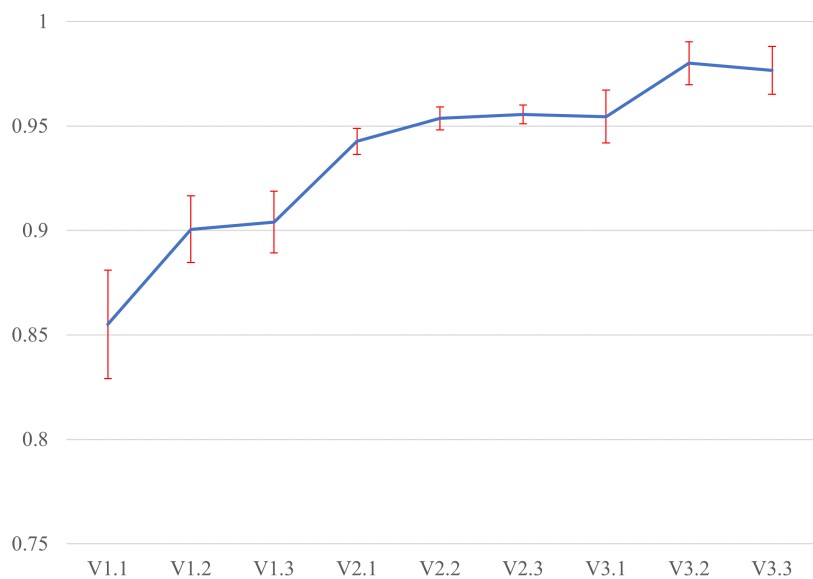

**Figure 4.** Comparison chart of the performance of variations.

Finally, a new algorithm was developed to predict the breakdown voltage in cases other than those existing in our database. The algorithm receives data from user-defined inputs and calculates the output value based on the defined neural network configuration. The graphics of Figures A10–A14 were generated using the algorithm that allows for the visualization of the relationship of the properties used with the tensile strength.

The graphs generated by the algorithm showed an acceptable behavior. The extremities presented a certain variation, which was something predicted, due to the amount of data being more centralized, causing uncertainties in the points with less data.

Grounded on the graphs, the parameter with the most decisive influence on increasing the tensile strength is the filling density, presenting an almost linear and increasing behavior with an increasing density since a more significant amount of material would be sharing the load supported by the piece.

The second parameter that most affects the attraction resistance is the filling orientation. This fact is linked to the mechanics of the material. A combination of components of everyday stresses with shear stresses between the layers of material justifies this behavior.

Compared with other parameters, the thickness did not affect the tensile strength, presenting an almost constant behavior throughout the analyzed thickness range.

## 5. Conclusions and Future Perspectives

Artificial intelligence can be helpful in several areas of activity. We presented a model that uses ANNs and a genetic algorithm (neuroevolutionary approach) to estimate the tensile strength of manufactured parts made by 3D printing. Therefore, for this paper, we also generated a database from experimental literature that investigated the effect of process parameters and the result obtained for the tensile strength property of the material. Then, an ANN model was developed for the own database, which predicts the tensile strength as a function of the printing process parameters. The developed model can be helpful in structural and economic optimizations of parts that perform an engineering function.

The execution plan of the neuroevolutionary model showed a difference between the values of $R^2$, which happens due to the random processes of the genetic algorithm, such as the generation of the initial population, selection, crossover and mutation. We verified that applying a population metaheuristic with genetic algorithms leads to potential gains in learning a neural network and confers an outstanding speed in the training mechanism. The idea of feedforward and backpropagation can be pretty compelling, assuming that they minimize the quadratic error. However, performing the training function for specific applications takes a long time.

In future research, we intend to replace the evolutionary aspect of our learning model with one that tends to converge more quickly. In our planning, we will apply the training method using a variety of genetic algorithms known as biased random-key genetic algorithms [35,36]. The primary purpose is to consider larger populations to achieve higher learning rates, assuming that the convergence time of the solutions tends to be smaller. Thus, we can further diversify the execution parameters of the evolutionary algorithm by establishing similar execution time limits to define the model.

**Author Contributions:** Conceptualization, R.R.B.M.; data curation, M.A.d.S. and R.R.B.M.; methodology, B.A.J. and P.R.P.; software, M.A.d.S.; validation, B.A.J.; writing—original draft, B.A.J. and P.R.P.; writing—review and editing, B.A.J. and M.A.d.S. All authors have read and agreed to the published version of the manuscript.

**Funding:** This research received no external funding.

**Institutional Review Board Statement:** Not applicable.

**Informed Consent Statement:** Not applicable.

**Data Availability Statement:** Publicly available datasets were analyzed in this study. These data can be found here: https://github.com/matheusalencar23/tcc, accessed on 17 May 2022. The README.md file talks a little about the repository. The "data.csv" file contains the data used for training. The "main.py" file contains the algorithm responsible for the neural network optimization tests based on the genetic algorithm. The "test.py" file contains the algorithm used to generate the loss curve graphs. The "af.py" file is responsible for the final implementation of the optimized network. The "helpers.py" file contains functions used in different situations. The "images" folder and the files "tests.txt", "times.txt" and "data_table.csv" contain the results of the optimizations. The results are separated into "v1", "v2" and "v3" folders based on the run settings.

**Acknowledgments:** Plácido Rogério Pinheiro is grateful to the University of Fortaleza/Edson Queiroz Foundation and to the National Council for Scientific and Technological Development (CNPq) for developing this project.

**Conflicts of Interest:** The authors declare no conflict of interest.

## Abbreviations

The following abbreviations are used in this manuscript:

| | |
|---|---|
| ABS | Acrylonitrile Butadiene Styrene |
| AI | Artificial Intelligence |
| ANN | Artificial Neural Network |
| FFF | Fused Filament Fabrication |
| GA | Genetic Algorithm |
| MLP | Multilayer Perceptron |
| PA | Polyamide |
| PC | Polycarbonate |
| PLA | Polylactic Acid |
| PP | Polypropylene |

## Appendix A

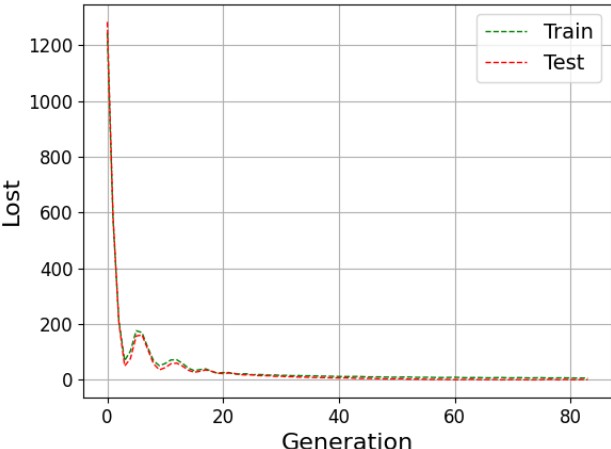

**Figure A1.** Variation 1.1 anomaly test.

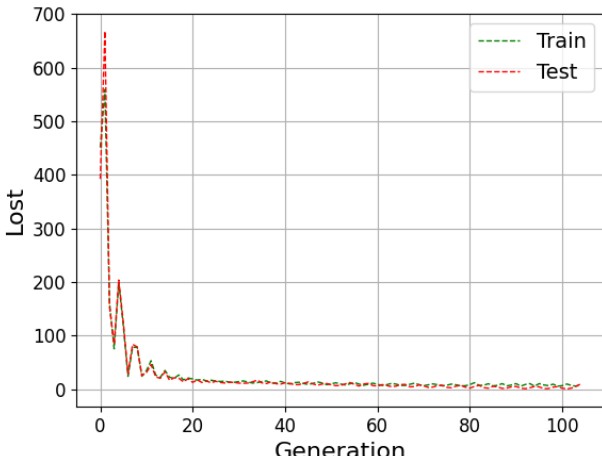

**Figure A2.** Variation 1.2 anomaly test.

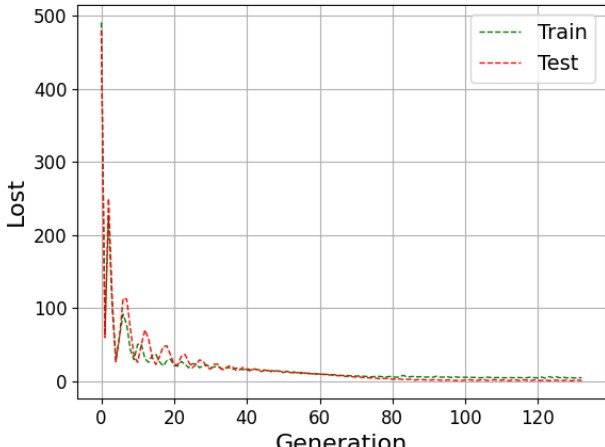

**Figure A3.** Variation 1.3 anomaly test.

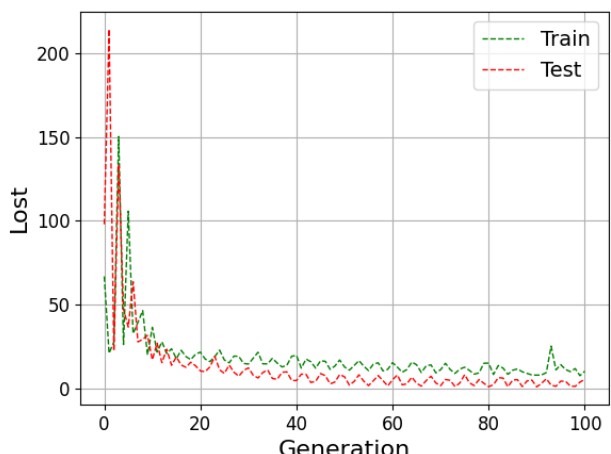

**Figure A4.** Variation 2.1 anomaly test.

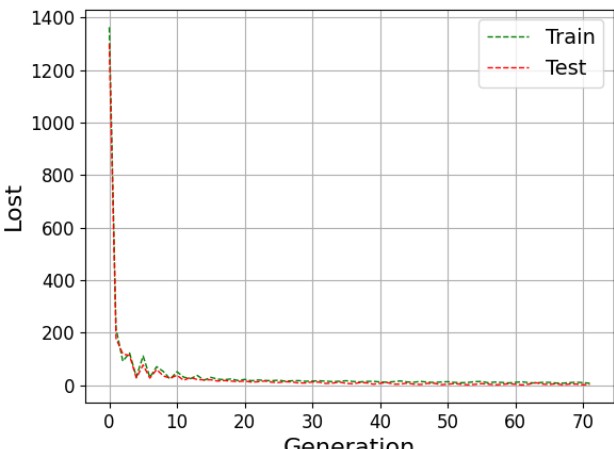

**Figure A5.** Variation 2.2 anomaly test.

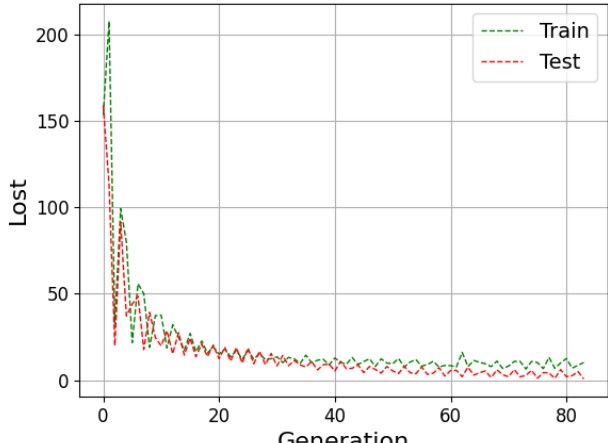

**Figure A6.** Variation 2.3 anomaly test.

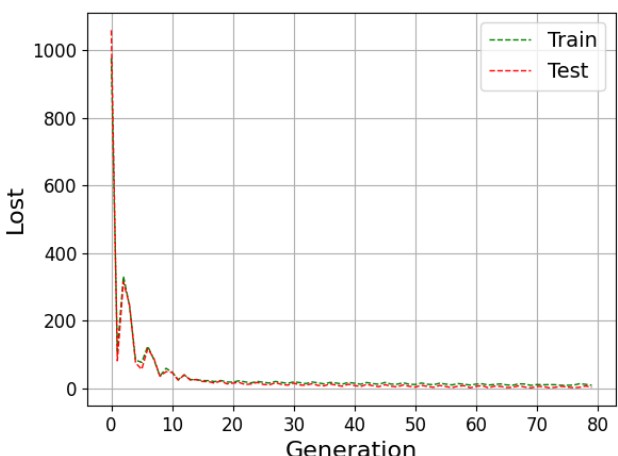

**Figure A7.** Variation 3.1 anomaly test.

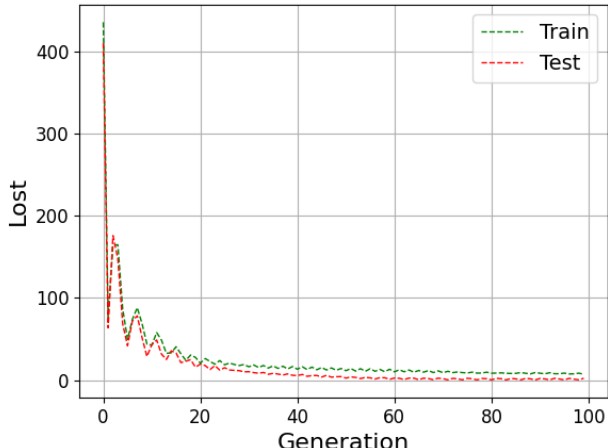

**Figure A8.** Variation 3.2 anomaly test.

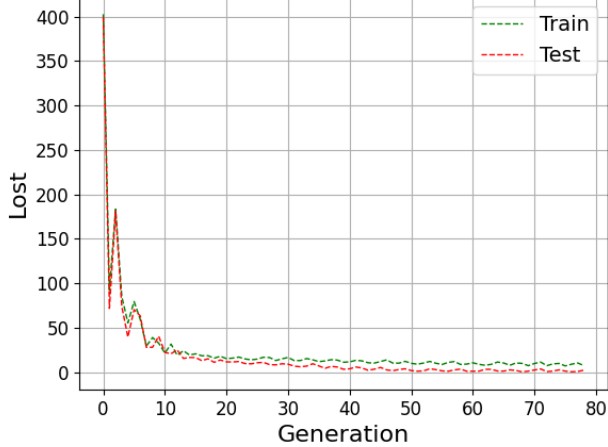

**Figure A9.** Variation 3.3 anomaly test.

**Table A1.** Used database.

| Work | Print Speed (mm/s) | Extrusion Temperature (°C) | Fill Density (%) | Thickness (mm) | Orientation (°) | Tensile Strength (MPa) |
|---|---|---|---|---|---|---|
| 1 | 40 | 190 | 100 | 0.2 | 0 | 40.03 |
| 1 | 40 | 210 | 100 | 0.2 | 0 | 50 |
| 1 | 40 | 230 | 100 | 0.2 | 0 | 56.96 |
| 1 | 50 | 190 | 100 | 0.2 | 0 | 50.04 |
| 1 | 50 | 210 | 100 | 0.2 | 0 | 59.84 |
| 1 | 50 | 230 | 100 | 0.2 | 0 | 49.86 |
| 2 | 30 | 200 | 100 | 0.1 | 90 | 45.83 |
| 2 | 30 | 200 | 100 | 0.1 | 90 | 50.35 |
| 2 | 30 | 200 | 100 | 0.1 | 90 | 47.83 |
| 2 | 30 | 210 | 100 | 0.1 | 90 | 47.52 |
| 2 | 30 | 210 | 100 | 0.1 | 90 | 50.72 |
| 2 | 30 | 210 | 100 | 0.1 | 90 | 50.08 |
| 2 | 30 | 220 | 100 | 0.1 | 90 | 48.34 |
| 2 | 30 | 220 | 100 | 0.1 | 90 | 51.38 |
| 2 | 30 | 220 | 100 | 0.1 | 90 | 49.21 |
| 2 | 30 | 230 | 100 | 0.1 | 90 | 49.35 |
| 2 | 30 | 230 | 100 | 0.1 | 90 | 50.68 |
| 2 | 30 | 230 | 100 | 0.1 | 90 | 50.46 |
| 2 | 30 | 240 | 100 | 0.1 | 90 | 49.32 |
| 2 | 30 | 240 | 100 | 0.1 | 90 | 50.13 |
| 2 | 30 | 240 | 100 | 0.1 | 90 | 49.52 |
| 2 | 40 | 200 | 100 | 0.1 | 90 | 50.26 |
| 2 | 40 | 200 | 100 | 0.1 | 90 | 49.62 |
| 2 | 40 | 200 | 100 | 0.1 | 90 | 49.05 |
| 2 | 50 | 200 | 100 | 0.1 | 90 | 50.54 |
| 2 | 50 | 200 | 100 | 0.1 | 90 | 50.23 |
| 2 | 50 | 200 | 100 | 0.1 | 90 | 49.41 |
| 2 | 60 | 200 | 100 | 0.1 | 90 | 52.15 |
| 2 | 60 | 200 | 100 | 0.1 | 90 | 51.73 |
| 2 | 60 | 200 | 100 | 0.1 | 90 | 50.51 |
| 3 | 60 | 210 | 100 | 0.1 | 0 | 27.48 |
| 3 | 60 | 210 | 100 | 0.1 | 15 | 30.69 |
| 3 | 60 | 210 | 100 | 0.1 | 30 | 32.35 |
| 3 | 60 | 210 | 100 | 0.1 | 45 | 37.42 |
| 3 | 60 | 210 | 100 | 0.1 | 60 | 43.93 |
| 3 | 60 | 210 | 100 | 0.1 | 75 | 49.85 |
| 3 | 60 | 210 | 100 | 0.1 | 90 | 53.66 |
| 3 | 60 | 210 | 100 | 0.05 | 90 | 53.7 |
| 3 | 60 | 210 | 100 | 0.15 | 90 | 51.75 |
| 3 | 60 | 210 | 100 | 0.2 | 90 | 50.52 |
| 3 | 60 | 210 | 20 | 0.1 | 90 | 20.04 |
| 3 | 60 | 210 | 40 | 0.1 | 90 | 21.08 |
| 3 | 60 | 210 | 60 | 0.1 | 90 | 23.81 |
| 3 | 60 | 210 | 80 | 0.1 | 90 | 28.5 |
| 3 | 60 | 195 | 100 | 0.1 | 90 | 46.97 |
| 3 | 60 | 200 | 100 | 0.1 | 90 | 47.3 |
| 3 | 60 | 205 | 100 | 0.1 | 90 | 49.18 |
| 3 | 60 | 215 | 100 | 0.1 | 90 | 54.39 |
| 3 | 60 | 220 | 100 | 0.1 | 90 | 54.17 |
| 3 | 60 | 225 | 100 | 0.1 | 90 | 54.27 |
| 3 | 60 | 230 | 100 | 0.1 | 90 | 53.03 |

**Table A1.** *Cont.*

| Work | Print Speed (mm/s) | Extrusion Temperature (°C) | Fill Density (%) | Thickness (mm) | Orientation (°) | Tensile Strength (MPa) |
|------|------|------|------|------|------|------|
| 4 | 80 | 200 | 100 | 0.2 | 90 | 38.43 |
| 4 | 80 | 200 | 100 | 0.2 | 90 | 37.69 |
| 4 | 80 | 200 | 100 | 0.2 | 90 | 35.78 |
| 4 | 80 | 200 | 100 | 0.2 | 90 | 37.61 |
| 4 | 80 | 200 | 100 | 0.2 | 90 | 37.71 |
| 4 | 80 | 200 | 100 | 0.2 | 90 | 36.5 |
| 4 | 80 | 200 | 100 | 0.2 | 90 | 36.41 |
| 4 | 80 | 200 | 100 | 0.2 | 90 | 38.12 |
| 4 | 80 | 200 | 100 | 0.2 | 90 | 37.33 |
| 4 | 80 | 200 | 100 | 0.2 | 90 | 35.58 |
| 4 | 80 | 200 | 100 | 0.2 | 90 | 36.53 |
| 4 | 80 | 200 | 100 | 0.2 | 90 | 36.69 |
| 4 | 80 | 200 | 100 | 0.2 | 90 | 39.06 |
| 4 | 80 | 200 | 100 | 0.2 | 90 | 39.15 |
| 4 | 80 | 200 | 100 | 0.2 | 90 | 39.15 |
| 5 | 60 | 215 | 100 | 0.1 | 0 | 28.67 |
| 5 | 60 | 215 | 100 | 0.1 | 0 | 25.07 |
| 5 | 60 | 215 | 100 | 0.1 | 0 | 26.21 |
| 5 | 60 | 215 | 100 | 0.1 | 0 | 27.66 |
| 5 | 60 | 215 | 100 | 0.1 | 45 | 30.84 |
| 5 | 60 | 215 | 100 | 0.1 | 45 | 32.97 |
| 5 | 60 | 215 | 100 | 0.1 | 45 | 32.94 |
| 5 | 60 | 215 | 100 | 0.1 | 45 | 28.56 |
| 5 | 60 | 215 | 100 | 0.1 | 90 | 54.37 |
| 5 | 60 | 215 | 100 | 0.1 | 90 | 55.97 |
| 5 | 60 | 215 | 100 | 0.1 | 90 | 57.24 |
| 5 | 60 | 215 | 100 | 0.2 | 0 | 25.53 |
| 5 | 60 | 215 | 100 | 0.2 | 0 | 24.95 |
| 5 | 60 | 215 | 100 | 0.2 | 0 | 26.2 |
| 5 | 60 | 215 | 100 | 0.2 | 0 | 23.05 |
| 5 | 60 | 215 | 100 | 0.2 | 45 | 31.47 |
| 5 | 60 | 215 | 100 | 0.2 | 45 | 30.56 |
| 5 | 60 | 215 | 100 | 0.2 | 45 | 30.02 |
| 5 | 60 | 215 | 100 | 0.2 | 90 | 51.18 |
| 5 | 60 | 215 | 100 | 0.2 | 90 | 53.53 |
| 5 | 60 | 215 | 100 | 0.2 | 90 | 54.53 |
| 5 | 60 | 215 | 100 | 0.2 | 90 | 57.65 |
| 5 | 60 | 215 | 100 | 0.3 | 0 | 23.56 |
| 5 | 60 | 215 | 100 | 0.3 | 0 | 24.14 |
| 5 | 60 | 215 | 100 | 0.3 | 0 | 23.63 |
| 5 | 60 | 215 | 100 | 0.3 | 45 | 29.32 |
| 5 | 60 | 215 | 100 | 0.3 | 45 | 29.19 |
| 5 | 60 | 215 | 100 | 0.3 | 45 | 28.98 |
| 5 | 60 | 215 | 100 | 0.3 | 90 | 44.94 |
| 5 | 60 | 215 | 100 | 0.3 | 90 | 45.62 |
| 5 | 60 | 215 | 100 | 0.3 | 90 | 45.24 |
| 5 | 60 | 215 | 100 | 0.3 | 90 | 48.71 |

**Table A1.** *Cont.*

| Work | Print Speed (mm/s) | Extrusion Temperature (°C) | Fill Density (%) | Thickness (mm) | Orientation (°) | Tensile Strength (MPa) |
|---|---|---|---|---|---|---|
| 6 | 80 | 200 | 20 | 0.1 | 90 | 20.71 |
| 6 | 80 | 200 | 20 | 0.2 | 90 | 19.09 |
| 6 | 80 | 200 | 20 | 0.1 | 45 | 20.2 |
| 6 | 80 | 200 | 20 | 0.2 | 45 | 17.67 |
| 6 | 80 | 220 | 20 | 0.1 | 90 | 22.63 |
| 6 | 80 | 220 | 20 | 0.2 | 90 | 19.79 |
| 6 | 80 | 220 | 20 | 0.1 | 45 | 21.36 |
| 6 | 80 | 220 | 20 | 0.2 | 45 | 18.2 |
| 6 | 80 | 200 | 40 | 0.1 | 90 | 24.18 |
| 6 | 80 | 200 | 40 | 0.2 | 90 | 22.35 |
| 6 | 80 | 200 | 40 | 0.1 | 45 | 19.91 |
| 6 | 80 | 200 | 40 | 0.2 | 45 | 22.24 |
| 6 | 80 | 220 | 40 | 0.1 | 90 | 24.97 |
| 6 | 80 | 220 | 40 | 0.2 | 90 | 26.14 |
| 6 | 80 | 220 | 40 | 0.1 | 45 | 25.31 |
| 6 | 80 | 220 | 40 | 0.2 | 45 | 24.32 |
| 6 | 80 | 200 | 60 | 0.1 | 90 | 26.23 |
| 6 | 80 | 200 | 60 | 0.2 | 90 | 26.55 |
| 6 | 80 | 200 | 60 | 0.1 | 45 | 29.43 |
| 6 | 80 | 200 | 60 | 0.2 | 45 | 25.22 |
| 6 | 80 | 220 | 60 | 0.1 | 90 | 30.22 |
| 6 | 80 | 220 | 60 | 0.2 | 90 | 28.67 |
| 6 | 80 | 220 | 60 | 0.1 | 45 | 29.43 |
| 6 | 80 | 220 | 60 | 0.2 | 45 | 26.71 |
| 7 | 30 | 185 | 10 | 0.15 | 90 | 18.6 |
| 7 | 30 | 185 | 10 | 0.19 | 90 | 22.79 |
| 7 | 30 | 185 | 10 | 0.25 | 90 | 25.16 |
| 7 | 30 | 185 | 10 | 0.15 | 90 | 21.4 |
| 7 | 30 | 185 | 10 | 0.19 | 90 | 23.63 |
| 7 | 30 | 185 | 10 | 0.25 | 90 | 29.15 |
| 7 | 30 | 185 | 10 | 0.15 | 90 | 25.76 |
| 7 | 30 | 185 | 10 | 0.19 | 90 | 24.6 |
| 7 | 30 | 185 | 10 | 0.25 | 90 | 30.38 |
| 7 | 30 | 185 | 25 | 0.15 | 90 | 22.04 |
| 7 | 30 | 185 | 25 | 0.19 | 90 | 26.24 |
| 7 | 30 | 185 | 25 | 0.25 | 90 | 30.71 |
| 7 | 30 | 185 | 25 | 0.15 | 90 | 26.53 |
| 7 | 30 | 185 | 25 | 0.19 | 90 | 28.8 |
| 7 | 30 | 185 | 25 | 0.25 | 90 | 32.38 |
| 7 | 30 | 185 | 25 | 0.15 | 90 | 30.33 |
| 7 | 30 | 185 | 25 | 0.19 | 90 | 34.16 |
| 7 | 30 | 185 | 25 | 0.25 | 90 | 36.37 |
| 7 | 30 | 185 | 50 | 0.15 | 90 | 30.93 |
| 7 | 30 | 185 | 50 | 0.19 | 90 | 35.47 |
| 7 | 30 | 185 | 50 | 0.25 | 90 | 35.54 |
| 7 | 30 | 185 | 50 | 0.15 | 90 | 30.79 |
| 7 | 30 | 185 | 50 | 0.19 | 90 | 34.05 |
| 7 | 30 | 185 | 50 | 0.25 | 90 | 36.96 |
| 7 | 30 | 185 | 50 | 0.15 | 90 | 30.26 |
| 7 | 30 | 185 | 50 | 0.19 | 90 | 35.78 |
| 7 | 30 | 185 | 50 | 0.25 | 90 | 37.7 |

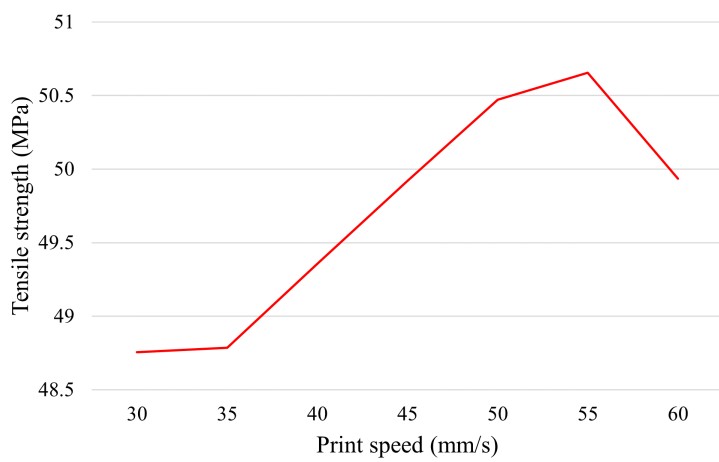

**Figure A10.** Graphs tensile strength vs. print speed generated from the algorithm.

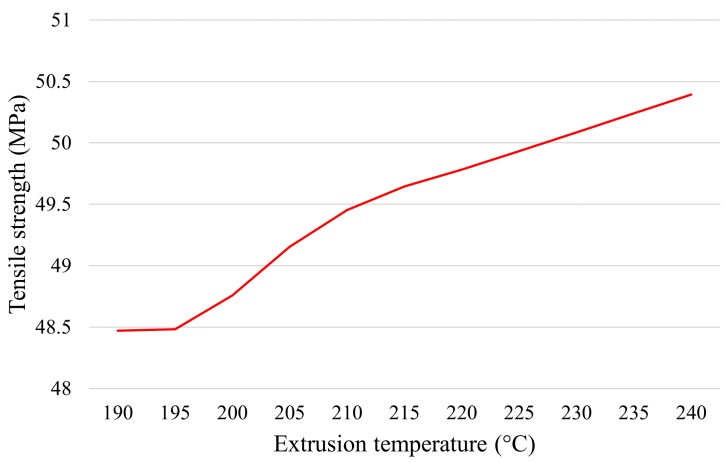

**Figure A11.** Graphs tensile strength vs. extrusion temperature generated from the algorithm.

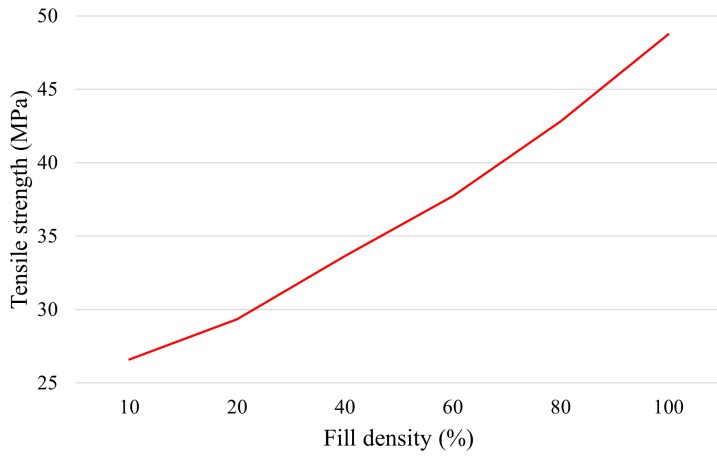

**Figure A12.** Graphs tensile strength vs. fill density generated from the algorithm.

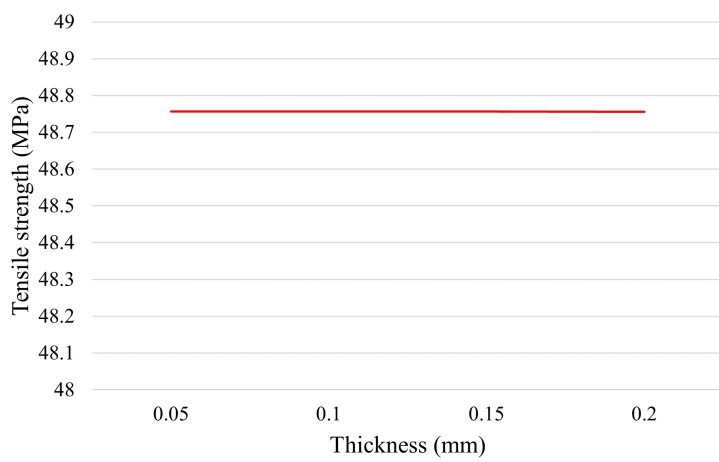

**Figure A13.** Graphs tensile strength vs. thickness generated from the algorithm.

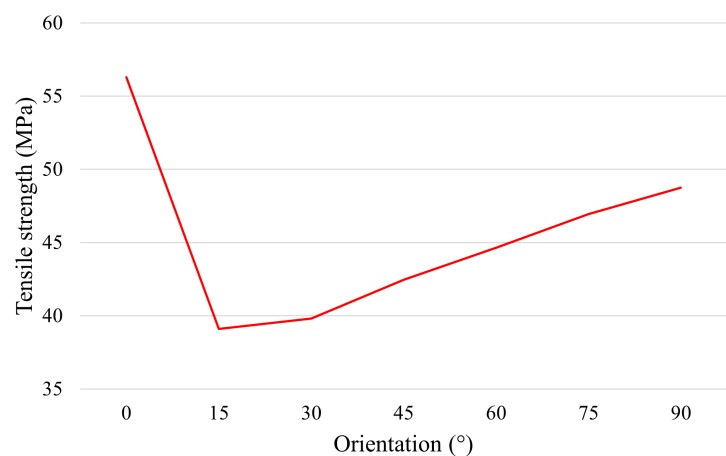

**Figure A14.** Graphs tensile strength vs. orientation generated from the algorithm.

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
