# Peer review of "A Neuroevolutionary Model to Estimate the Tensile Strength of Manufactured Parts Made by 3D Printing"

_algorithms, doi:10.3390/a15080263_

Round 1
Reviewer 1 Report
Use of FFF (Fused Filament Fabrication) is suggested. FDM is a commercial name.
You use 3d printing and 3D printing. Just use 3D Printing.
The term ‘fusion and deposition modelling’ is wrong. Use just FFF.
53: After defining ANN, stop writing the whole name. Use just ANN.
54: needs a reference.
Figure 1 is not a good representation. Better one is needed.
Figure 2 and 3 are not yours. Their impact is low to your research. Why do you add them into your paper?
MLP needs a better definition and a reference.
166 is an equation.
GA was defined before. Why do you define it again in 176.
299: define the CSV files.
Figure 8 has no impact on the paper.
The conclusion is not acceptable. It is not factual. Major cuts are needed. Very short and factual conclusion is required.
Several references are redundant. No need to have them since their processes are not used in this paper. Examples are 7, 9, 25
Author Response
Dear reviewer, we would like to thank you for taking the time and effort to improve this article. The suggested revision is provided below.
Sincerely,
The authors
Unifor-Brazil
Reviewer’s Comments for the author “A neuroevolutionary model to estimate the tensile strength of manufactured parts made by 3D printing”
Comments and Suggestions for Authors
We appreciate all comments and suggestions and present, below, the answers about the corrections.
Item1: Use of FFF (Fused Filament Fabrication) is suggested. FDM is a commercial name.
Response: Reviwed. We change the initials FDM to FFF.
Item 2: You use 3d printing and 3D printing. Just use 3D Printing.
Response: Reviwed. We change it in the final document.
Item 3: The term ‘fusion and deposition modelling’ is wrong. Use just FFF.
Response: Reviwed. We change it in the final document.
Item 4: 53: After defining ANN, stop writing the whole name. Use just ANN.
Response: Reviwed. We change it in the final document.
Item 5: 54: needs a reference.
Response: Reviwed. Reference entered: 7. Kant, A.; Suman, P.K.; Giri, B.K.; Tiwari, M.K.; Chatterjee, C.; Nayak, P.C.; Kumar, S. Comparison of multi-objective evolutionary neural network, adaptive neuro fuzzy inference system and bootstrap-based neural network for flood forecasting. Neural Comput 407 & Applic 2013, 23, 231–246. doi:https://doi.org/10.1007/s00521-013-1344-8.
Item 6: Figure 1 is not a good representation. Better one is needed.
Response: Reviwed. We change the figure considering a better representation.
Item 7: Figure 2 and 3 are not yours. Their impact is low to your research. Why do you add them into your paper?
Response: Figures 2 and 3 were added by request of insertion in the previous review (by the editor).spi
Item 8: MLP needs a better definition and a reference.
Response: Reviwed. We redefine the MLP definition and add reference [16] Zare, M.; Pourghasemi, H.R.; Vafakhah, M.; Pradhan, B. Landslide susceptibility mapping at Vaz Watershed (Iran) using an
artificial neural network model: a comparison between multilayer perceptron (MLP) and radial basic
function (RBF) algorithms 2013. 6, 2873–2888. doi:https://doi.org/10.1007/s12517-012-0610-x.
Item 9: 166 is an equation.
Response: O(f(n)) is an asymptotic notation. It's a runtime function that can happen in the worst
input situations (i.e. everything) on the parameters mentioned: t, f, h, n and o.
We add this text to try to explain "Note that Big-O quota is a mathematical notation that describes
the limiting behavior of a function when the argument tends towards a particular value or infinity."
Item 10: GA was defined before. Why do you define it again in 176.
Response: In the introduction, we briefly commented on GA. However, we thought it necessarily
better to describe GA in the specific topic about GA.
Item 11: 299: define the CSV files.
Response: These are (CSV - Comma Separated Values). Now, we inform it in the text.
.
Item 12: Figure 8 has no impact on the paper.
Response: Reviwed. Figure 8 removed.
Item 13: The conclusion is not acceptable. It is not factual. Major cuts are needed. Very short and
factual conclusion is required.
Response: Reviwed. We change the text.
Item 14: Several references are redundant. No need to have them since their processes are not
used in this paper. Examples are 7, 9, 25.
Response: Reviwed. We removed references 7 and 9, but we consider that Reference 25 (from the
submission version) cannot be excluded. Part of the data used to build the database was taken from
this reference.

Reviewer 2 Report
I have reviewed the paper “A neuroevolutionary model to estimate the tensile strength of manufactured parts made by 3D printing” and I would like to congratulate the authors for their work. In general, the work is interesting, but I have several major concerns that need to be addressed from the authors. Therefore, I would like from the authors to elaborate the next comments:
1)
Paragraph 2.4 is not necessary in the paper since it is trivia knowledge. You can refer in two-three sentences for underfit and overfit.
2)
How did you conclude to examine the following parameters for evaluating the mechanical performance of a 3D printed part? (Printing speed, extrusion temperature, fill density, extruded filament thickness and extrusion orientation).
3)
Is the size of your dataset enough in order to develop reliable models?
4)
I think more description of the applied model in section 3 is necessary.
5)
double dots in line 328 at page 10. Please check it.
6)
You should insert text between figures or tables. It is not preferable to have table 5, then figure 7 and then table 6 without the intervention of text supporting these data.
7)
3D plots of tensile strength versus specific parameters is preferable instead of figure 8.
8)
Which investigated parameters has the highest impact on the tensile strength?
9)
I think a figure presenting the applied experimental process will be valuable.
Author Response
Dear reviewer, we would like to thank you for taking the time and effort to improve this article. The suggested revision is provided below.
Sincerely,
The authors
Unifor-Brazil
Reviewer’s Comments for the author “A neuroevolutionary model to estimate the tensile strength of manufactured parts made by 3D printing”
Comments and Suggestions for Authors
I have reviewed the paper “A neuroevolutionary model to estimate the tensile strength of manufactured parts made by 3D printing” and I would like to congratulate the authors for their work. In general, the work is interesting, but I have several major concerns that need to be addressed from the authors. Therefore, I would like from the authors to elaborate the next comments:
We appreciate all comments and suggestions and present, below, the answers about the corrections.
1) Paragraph 2.4 is not necessary in the paper since it is trivia knowledge. You can refer in two-three sentences for underfit and overfit.
Response: Reviwed. We remove section 2.4 and change the local of the underfit and overfit concepts for section 2.2.
2) How did you conclude to examine the following parameters for evaluating the mechanical performance of a 3D printed part? (Printing speed, extrusion temperature, fill density, extruded filament thickness and extrusion orientation).
Response: The mentioned parameters are the primary (or usual) parameters for FFF 3D printing. These are the parameters used in experimental research in which we consider creating the input database.
3) Is the size of your dataset enough in order to develop reliable models?
Response: We consider seven experimental studies developed by other authors to construct the input data. They were allowing a good range of main 3D printing FFF process variables. The results show that the model can predict the properties well, especially in the central region of the content of variables, where values are more common in the 3D printing of PLA. A vast database would generate a model without much use since all the material's behaviour would be known without the need for a model to predict it.
4) I think more description of the applied model in section 3 is necessary.
Response: A model built by a neural network is a hypothesis that tries to achieve a real input and output function. We believe the paper describes all the essential steps for replicating our model and exposes the input composition, the layers, the output and the learning process through the GA, between 254 and 300. We apologize, but could the reviewer distinguish which point needs a better explanation?
5) double dots in line 328 at page 10. Please check it.
Response: Reviwed.
6) You should insert text between figures or tables. It is not preferable to have table 5, then figure 7
and then table 6 without the intervention of text supporting these data.
Response: Reviwed. We change the text considering this advice.
7) 3D plots of tensile strength versus specific parameters is preferable instead of figure 8.
Response: The other reviewer requested the deletion of figure 8. Therefore, we chose to remove
this figure. We hope that there is no loss for the text.
8) Which investigated parameters has the highest impact on the tensile strength?
Response: Reviwed. We changed the text and added this information in 337 to 348.
9) I think a figure presenting the applied experimental process will be valuable.
Response: Data were not obtained experimentally by the authors of the paper. We get all input
information from experimental studies developed by other authors, as described in 262 to 268 in
Table 1.

Reviewer 3 Report
The aim of this paper is to present a neuroevolutionary model to estimate the tensile strength of manufactured parts made with FFF and PLA material.
The study of the application of artificial intelligence algorithms to the FFF manufacturing process is nowadays an interesting subject from a scientific and industrial point of view, however, unfortunately, the research developed by the authors presents certain limitations
1.- The authors present a new neuroevolutionary model obtained by combining the use of neural networks with genetic-type evolutionary algorithms. Genetic algorithms applied to manufacturing systems usually use the equations of the physical model of the system together with a set of boundary conditions and parameters that allow obtaining the optimal values in compliance with the equations of the physical model.
However, the model presented by the authors does not make use of any physical model for system optimization. Authors are recommended to detail this topic.
2.- The authors present in the attached item of the paper a set of representative graphs of the variation of the tensile strength against a series of manufacturing parameters. However, they do not present the application of the results to a real industrial case together with an experimental validation that allows evaluating the precision of the presented algorithm.
Authors are recommended to validate their research on specific and real manufacturing cases where the accuracy of the algorithm developed is clear by comparing the resulting values of the algorithm with experimental values.
Author Response
Dear reviewer, we would like to thank you for taking the time and effort to improve this article. The suggested revision is provided below.
Sincerely,
The authors
Unifor-UFC-Brazil

Reviewer 4 Report
Dear Authors,
the work described should be improved before the publication, in particular the background should be better described.
The text denotes a performing programming work but the key concepts are exposed with little clarity, as if they had been extracted with a low awareness of the topics at a theoretical level.
Line 38 - ANN is a computational model (ISBN: 978-0-12-818438-7)
Line 40 - NNs can be trained using both supervised and unsupervised learning
Lines 82/84 - computer model? Improve: CAD, CAM, Firmware (roles, differences, ..)
Line 86 - "3D printing uses low-cost materials and generates little waste"
There are 7 families of additive manufacturing technologies….
Please describe in more detail.
Line 87 - FFF is the most common (Are you sure? And SLA?)
Please add some quotes
Bullets from Line 98 - Add build platform temperature, ambient temperature and humidity (not considered in your NNs, but useful for future studies) ... (https://www.mdpi.com/2073- 4360/13/4/617 / htm)
Figures 2, 3 and 4 - Please describe the correlation with the mathematical functions (delete figures)
Please add the Statistical Analysis paragraph
Author Response

(The authors gave the same response as above.)

Round 2
Reviewer 1 Report
Revisions are acceptable.
Author Response

(The authors gave the same response as above.)

Reviewer 2 Report
I reviewed the revised version of the paper “A neuroevolutionary model to estimate the tensile strength of manufactured parts made by 3D printing”. In general, the work is weak in the content and the accomplished results. The research topic is interesting, but in the present work is based on experimental results from seven other studies, where various external parameters could affect the results. In addition, the size of the experimental set is relatively small.
Author Response

(The authors gave the same response as above.)

Reviewer 3 Report
The authors have addressed satisfactorily the points raised during the review
Author Response
Dear reviewer, we would like to thank you for taking the time and effort to improve this article. The suggested revision is provided below.
Sincerely,
The authors

Reviewer 4 Report
Dear Authors,
in my opinion, the document could be accepted in the present form.
Author Response

(The authors gave the same response as above.)

Round 3
Reviewer 2 Report
The research topic is interesting, but in the present work is based on experimental results from other studies, where various external parameters could affect the results. In addition, the size of the experimental set is relatively small.
Author Response

(The authors gave the same response as above.)
